# A Platform Approach to Protein Encapsulates with Controllable Surface Chemistry

**DOI:** 10.3390/molecules27072197

**Published:** 2022-03-28

**Authors:** Nina Warner, Ilja Gasan Osojnik Črnivec, Vijay Kumar Rana, Menandro Cruz, Oren A. Scherman

**Affiliations:** 1Melville Laboratory for Polymer Synthesis, Yusuf Hamied Department of Chemistry, University of Cambridge, Lensfield Road, Cambridge CB2 1EW, UK; ncw39@cam.ac.uk (N.W.); vkr23@cam.ac.uk (V.K.R.); mvc34@cam.ac.uk (M.C.); 2Biotechnical Faculty, University of Ljubljana, Jamnikarjeva 101, SI-1000 Ljubljana, Slovenia; gasan.osojnik@bf.uni-lj.si

**Keywords:** spray dry, protein, encapsulation, formulation, excipient, core-shell, EISA

## Abstract

The encapsulation of proteins into core-shell structures is a widely utilised strategy for controlling protein stability, delivery and release. Despite the recognised utility of these microstructures, however, core-shell fabrication routes are often too costly or poorly scalable to allow for industrial translation. Furthermore, many scalable routes rely upon emulsion-techniques implicating denaturing or environmentally harmful organic solvents. Herein, we investigate core-shell protein encapsulation through single-feed, aqueous spray drying: a cheap, industrially ubiquitous particle-formation technology in the absence of organic solvents. We show that an excipient’s preference for the surface of the spray dried particle is well-predicted by its hydrodynamic diameter (D${}_{h}$) under relevant feed buffer conditions (pH and ionic strength) and that the predictive power of D${}_{h}$ is improved when measured at the spray dryer outlet temperature compared to room temperature ($R^{2}$ = 0.64 vs. 0.59). Lastly, we leverage these findings to propose an adaptable design framework for fabricating core-shell protein encapsulates by single-feed aqueous spray drying.

## 1. Introduction

### 1.1. Core-Shell Particles for Protein Encapsulation

The encapsulation of proteins within amorphous dried particles has become a ubiquitous paradigm across the pharmaceutical and food industries. In comparison to bulk drying methods, particle forming technologies afford enhanced control over the end product with respect to bulk material homogeneity, protein release kinetics, aerosolisability, and handling properties (i.e., powder flow) [1,2,3,4].

Additionally, these technologies may be coupled with particle engineering methods to access a vast array of advanced particle structures. Core-shell structured particles are amongst the most desirable, particularly with regard to applications involving protein encapsulation. In these cases, biphasic segregation of formulation components enables incorporation of multi-functional materials that might otherwise compromise protein stability in the bulk phase. Protein core-shell particles typically consist of an inner protein/stabilising excipient ’core’ surrounded by an outer ’shell’ layer, often comprised of a polymer or wax, which forms a protective encasing of the labile cargo. Core-shell structures have been used in protein formulations to introduce advanced functionalities such as high precision controlled/triggered release, selective gas/solvent permeability, in vivo targeting capabilities, enhanced bio-absorption [5], improved dissolvability, reduced particle agglomeration, and increased stability in the presence of various stress vectors: humidity, heat, light, oxidants, etc. A diverse array of benefits associated with these formulations has made the large-scale fabrication of particle-based protein encapsulates with highly ordered, predictable morphologies an attractive industrial target, which is particularly sought for controlled performance of the particle core.

### 1.2. Evaporation-Induced Self Assembly (EISA) of Core Shell Particles by Spray Drying

Spray drying offers a powerful industrial scale toolbox for precise powder formation. Moisture is removed in a simple, one-step process wherein a feed solution comprised of a solvent, API, and excipients is continuously processed into a micronized powder. This is achieved by atomising the feed into small droplets, evaporating the solvent in a hot, aspirated chamber, and capturing the resulting particles in a cyclone device. Typically, the feed and the drying gas are introduced into the vertical drying chamber in co-current flow.

While its high cost efficiency, adaptability, scalability, and batch-to-batch reproducibility make spray drying especially well-suited to industrial application, the process often lacks precise control over size uniformity characteristic of lab-scale techniques such as microfluidics [6]. As such, accessing structured particles on an industrial scale is considerably more challenging, though not altogether impossible.

Attempts to fabricate particles with controlled core-shell morphology from spray drying can be broadly categorised by three approaches, (1) instrument modification, (2) operating conditions and (3) feed composition. The first of these, instrument modification, refers to the re-configuration of spray dryer hardware, with successful applications encompassing developments such as ultrasonic and coaxial spray drying [7]; however, these can prove costly as well as difficult to upscale. The second approach, the modification of spray conditions, concerns the fine-tuning of drying settings such as aspirator strength, flow rate, and inlet temperature. However, some feed components, particularly proteins, often restrict the process to narrow operating windows [8]. The third approach is the direct modification of feed composition and encompasses changes to the solvent, dissolved components, and their concentrations. It is the final approach that will serve as the focus for the remainder of this discussion.

To explain the relationship between feed solution properties and resultant particle morphology, there have emerged three theories as highlighted in Figure 1. These theories describe the evaporation-induced self assembly as governed by diffusion, water evaporation flux, and surface activity, respectively; it is the first of these that is studied herein. In the diffusion-governed scenario, evaporation at the droplet surface results in transient spikes in the solute concentration [9]. As the droplet surface becomes increasingly concentrated, a diffusion gradient is established, which drives the inward migration of solute species. The speed at which this migration occurs is governed by the solute diffusion coefficient, *D*, which can be mathematically related to the solute hydrodynamic radius by the Stokes–Einstein equation [10], expressed as,
(1)$$D=\frac{k_{B}\;T}{6\pi \;\mu \;r_{u}}$$
wherein *D* is the particle diffusion coefficient, $k_{B}$ is Boltzmann’s constant, *T* is the absolute temperature, μ is the particle mobility, and $r_{u}$ is the particle hydrodynamic radius. Solute species with larger hydrodynamic radii ($r_{u}$), and in turn slower diffusion coefficients (*D*), lag behind their smaller counterparts. As a result, the larger species more quickly reach their saturation limits and precipitate at the droplet-air interface, forming a particle shell enriched with the larger solute species [9,11,12,13,14]. In the water-evaporation flux and surface activity theories, in contrast, the surface becomes enriched with hydrophilic and surface active compounds, respectively [15,16].

These theories are not mutually exclusive and in fact, it is likely a combination of all three that dictate the final morphology of a spray dried particle. Moreover, it is important to note that the relative contribution of each predictor should not be regarded as a constant, but rather as a complex function of system conditions (e.g., drying speed) and the degree of variance encompassed by the system components. Nevertheless, these theories provide useful frameworks for rational design of self-assembled core-shell structures by spray drying. For example, Chen et al. reported the single-step assembly of highly uniform core-shell structures from an aqueous two component system consisting of common biocompatible excipients: nanoparticles of Eudragit RS (ethyl acrylate-methyl methacrylate copolymer and a low content of methacrylic acid ester with quaternary ammonium groups) and silica sol (hydrolysed tetraethyl orthosilicate) [17]. The final microparticles exhibited a core-shell morphology comprised of a silica shell and Eudragit RS core. The authors attributed this segregation to the disparity in component hydrodynamic diameters ($D_{h}$); the $D_{h}$ of hydrolysed TEOS and Eudragit RS were 1 and 120 nm, respectively. While this study was proceeded by a number of reports demonstrating enrichment of larger solutes on the surface of the spray-dried particle, it served as the first example of true core-shell particle formation from an aqueous single-feed spray drying set-up [11,12,13,14].

### 1.3. EISA of Core Shell Protein Encapsulates by Spray Drying

Despite these developments, EISA of core-shell particles by single-feed spray drying remains challenging for even simple binary systems; these challenges are exacerbated in complex formulations wherein the encapsulated ’core’ species is a metastable biomacromolecule with surfactant character, i.e., a protein. In fact, the preferential migration of proteins to the droplet-air interface during the drying process makes them common choices for shell-forming encapsulation agents [18].

Nonetheless, when proteins are the active compound, there are a number of proven approaches for limiting their surface adsorption in spray dried particles. The most common and simple of these is the incorporation of surfactants. Indeed Pinto et al. performed an analysis of literature trends in spray dried protein pharmaceuticals and found that 10% of all feed solutions incorporated a surfactant of some kind. Furthermore, of the four commercially approved spray dried protein pharmaceuticals, two incorporate surfactant excipients [19]. These additives, however, do not generally provide stabilisation alone and in fact may compromise long-term stability of the protein powder. It is therefore necessary to incorporate surfactants alongside additional stabilising excipients [19]. Moreover, this approach is unfit for applications wherein a functional shell is desired as the chemical properties of the particle surface are determined only by those of the surfactant.

Limiting the surface adsorption of costly protein pharmaceuticals during spray drying has also been achieved by the addition of ’sacrifical’ protein species. These protein excipients competitively adsorb at the droplet-air interface, thereby displacing the more precious protein [20]. This method was employed by 3% of all reports of spray dried protein pharmaceuticals within the past 30 years [19]. The difference in preference for two proteins to adsorb at the air-water interface remains small, as such, high loadings of the excipient protein tend to be required for near complete competitive displacement of the protein of interest, significantly increasing the overall cost of formulation [21]. Furthermore, it should be noted that the incorporation of protein excipients in spray dried formulations has been shown to drastically effect the bioavailability of the pharmaceutical, often detrimentally [22]. The benefits of reduced surface adsorption therefore must be weighed against potential drawbacks associated with each unique formulation scenario.

### 1.4. Study Aims

We aimed to develop a scalable platform for the fabrication of core-shell protein encapsulates by simple, single-feed spray drying (Figure 1). The principles of EISA were adapted to an industrially representative system. A semi-pure, commercially-relevant protein was used whilst organic solvents, expensive and/or toxic chemicals, and specialised spray drying equipment were avoided to maintain industrial relevance and translatability of our findings [9].

Our investigation was designed with the intention to relate readily-tunable feed solution parameters to the core-shell morphology of dried protein encapsulates. To achieve this, we applied a modified fractional factorial Design of Experiment (DoE) to a series of sixteen feed solutions investigating six factors. Feed solutions were systematically modified to isolate the effects of (1) pH, (2) ionic strength, (3) excipient $D_{h}$, (4) excipient surface functionality, (5) total dissolved solids, and (6) the ratio of excipient to protein. To enhance the tunability of our system, we worked exclusively with silica nanoparticle excipients. These nanoparticles could be readily altered in terms of size and surface functionality, enabling the effects of the excipient sterics ($D_{h}$) and electronics (polarity, charge, etc.) to be directly studied. Moreover, the true size and surface chemistry of the nanoparticle excipients could be compared to the effective $D_{h}$ and zeta potential observed within the buffered feed solution. Relating these values to the obtained morphology gave insight into the extent to which aqueous ’solvent engineering’ could influence particle microstructure.

Overall, our work assesses the feasibility of using principles of EISA to access core-shell microparticles in an industrially representative system: namely the single-feed, aqueous spray drying of a semi-pure protein. We identify parameters with high predictive power and show how these can be tuned to control the surface preference of excipients. Furthermore we discuss how these predictors can be manipulated from both ex and in situ approaches. Our work provides insight on how tunable morphologies can be accessed in sensitive systems such as those containing biologics. Finally, we propose a highly adaptable and simple platform approach to enhance the extent of encapsulation for a wider array of bioactive compounds.

## 2. Results and Discussion

### 2.1. System Design

A series of of twenty-two feed solutions were designed to investigate the effects of (1) pH, (2) ionic strength, (3) excipient $D_{h}$, (4) excipient surface functionality, (5) total dissolved solids, and (6) the ratio of excipient to protein on the morphology and surface composition of spray dried protein formulations (Table 1).

Of the formulations included, sixteen contained excipients—fourteen of these were studied by scanning electron microscopy (SEM), X-ray photoelectron spectroscopy (XPS), and elemental analysis (EA) to assess both the morphology and surface composition of the obtained particles. The two remaining formulations were designed to probe the influence of the total dissolved solids content and the ratio of excipient to protein on particle morphology (5-0[med-OH]${}^{15}$ and 5-0[med-OH]${}_{50}$). These were characterised only by SEM (Figure A6 and Figure A9). The five feed solutions containing only protein and buffer were included as controls to isolate the effect of the excipient itself on the morphology of the spray dried particles (Figure A1).

Nanoparticle size and surface functionality were studied to probe the effect of directly modifying excipient molecular size and hydrophilicity. In contrast, feed solution pH and salt (CaCl${}_{2}$) concentration were investigated as indirect methods of controlling the effective $D_{h}$ and colloidal stability of the excipient. Two additional factors unrelated to excipient properties (total concentration of dissolved solids in the feed buffer ([Excipient + Protein]) and excipient loading ratio ([Excipient]/[Excipient + Protein])) were also studied to understand their effect on particle morphology. Parameter levels—selected taking into consideration synthetic feasibility/commercial availability, protein stability, and industrial relevance—are discretely defined in Figure 2.

### 2.2. Characterisation of Synthesised Nanoparticles

Three sizes of silica nanoparticles were compared. Particles of $D_{h}$ = 16 ± 1 nm (small) and $D_{h}$ = 38 ± 1 nm (medium) were purchased as commercially available Ludox suspensions (Figure A1). A third particle size—97 ± 2 nm (large)—was synthesised by a seed-growth method using AS40 Ludox silica as the precursor seed (Figure A2).

### 2.3. Characterisation of Functionalised Nanoparticles

To test the influence of surface functionality on spray dried morphology and nanoparticle surface adsorption, nanoparticles with three different surface functionalities (hydroxyl (SiOH/Si$O^{-}$), aminopropyl, and octyl) were studied. Unfunctionalised Ludox silica nanoparticles (AS40) contained a hydroxyl surface. Aminopropyl and octyl functionalised silica nanoparticles were prepared by modification of Ludox (AS40) as described in the Methods section. Particle functionalisation was confirmed by zeta potential (ζ) measurements in MilliQ water (Figure A3). The negatively-charged (−40 mV) hydroxylated (SiOH/Si$O^{-}$) surface and positively-charged (+35 mV) aminopropyl functionalisation exhibited good colloidal stability. Functionalisation with octyl groups yielded a near neutral (+3 mV) zeta potential indicative of an uncharged surface.

Nanoparticle functionalisation also influenced particle size (Figure A4). Both surface functionalities induced nanoparticle aggregation. Aggregation was more extensive when particles were functionalised with aminopropyl moieties, likely indicative of electrostatic interactions between functionalised (postively-charged) and residual unfunctionalised (negatively-charged) surface domains.

### 2.4. Characterisation of Colloidal Feed Solution

Nanoparticle excipients were also characterised under feed buffer conditions prior to spray drying. The measured $D_{h}$ and ζ values indicated the effective in situ size of the excipient nanoparticles under relevant processing conditions. To more accurately simulate the conditions during particle formation, characterisation was performed at both room temperature (RT) and the mean spray dryer outlet temperature (70 °C). The intensity weighted $D_{h}$ and ζ for buffered excipients at RT and $T_{outlet}$ are tabulated in Table 2.

### 2.5. Characterisation of Spray Dried Particles

#### 2.5.1. General Morphology

The morphologies of spray dried particles were assessed by SEM. Whilst the extent of core-shell structure could not be observed from microstructure alone, several general trends were found to characterise the morphologies of the systems studied. First, it was found that buffer composition (i.e., ionic strength and pH modifying components) strongly governed particle morphology in the absence of excipient (Figure 3). In particular, high salt concentrations tended to induce needle-like crystal formation and particle fusion. Upon the incorporation of nanoparticle excipients, however, these morphological changes could be counteracted (Figure 3). Further, it was found that the nature of the excipient—i.e., nanoparticle size (Figure A7) and/or surface functionality (Figure A8)—did not significantly influence the morphology of the obtained particles. These results suggest that the counteractive effect of nanoparticle excipients on buffer-induced particle morphology perturbations is likely attributable to the ’dilution’ of buffer components in the dried particle, an effect largely indifferent to the chemical and physical properties of the excipient. Further characterisation of particle morphology is provided in the Section A.2 of the Supplementary Information.

#### 2.5.2. Core-Shell Structure

The extent to which obtained particles exhibited core-shell morphology was assessed by the the procedure described in Figure 4. The representation of protein and nanoparticle excipient were tracked by measuring the abundance sulphur and silicon elements, respectively. Bulk compositions were determined by elemental analysis, whilst surface compositions were measured via XPS.

From the measured surface and bulk compositions of sulphur (a proxy for protein) and silicon (a proxy for nanoparticle excipients), the percent of preferential surface adsorption expressed by the excipient and protein could be readily calculated. These calculations, as well as the raw elemental compositions for Si and S in the dried material bulk and surface are reported in Table 3.

### 2.6. Investigation of Predictive Parameters

Diffusion controlled self assembly has been shown to yield core-shell structures wherein the shell layer is formed by components with slow diffusion coefficients (*D*) and in turn, large hydrodynamic size ($D_{h}$) (Figure 1). To test whether we could harness this phenomenon to control the surface composition of spray-dried particles containing protein, we compared the $D_{h}$ of three sizes of nanoparticle excipients (5-0[small-OH], 5-0[med-OH], and 5-0[large-OH]) in water against the surface preference exhibited by these excipients (Figure 5). The trend in size did not follow the trend in surface preference for the three samples studied, although the difference in preference for 5-0[small-OH] and 5-0[med-OH] was relatively small (−82% vs. −90%) compared to that calculated for 5-0[large-OH] (−24%). This changed, however, when the $D_{h}$ of the excipient in the feed buffer was plotted against surface preference; in this case, the $D_{h}$ did predict the excipient surface preference. A plot of the $D_{h}$ in buffer vs. the excipient surface preference for all three formulations yielded a straight line with an $R^{2}$ = 0.999 (Figure 5).

Contrary to expectation, the surface preference of the excipient in formulations 5-0[small-OH], 5-0[med-OH], and 5-0[large-OH] was negative in all three cases. This could be the result of competition from buffer salts (sodium acetate) precipitating at the particle surface. To test this theory, the sodium content was measured for each surface and found indeed to be high (27, 24 and 6 wt% for samples 5-0[small-OH], 5-0[med-OH], and 5-0[large-OH] respectively).

We next decided to study more broadly the relationship between an excipient’s in situ $D_{h}$ and preferential adsorption at the particle surface. We plotted the in situ $D_{h}$ of excipients in thirteen formulations against the measured preferential surface adsorption (Figure 6a). The results indicated a moderate linear correlation ($R^{2}$ = 0.59). Interestingly, this correlation improved ($R^{2}$ = 0.64) when the $D_{h}$ was measured at the spray dryer outlet temperature ($T_{out}$) instead of room temperature (Figure 6b).

From these results, we may conclude two key findings. First, the $D_{h}$ of an excipient is moderately predictive of its preference for the droplet-air interface during drying, and in turn for the surface of the spray dried particle. Second, it is important to consider the properties of an excipient under in situ operating conditions (i.e., pH, ionic strength, temperature) as the $D_{h}$ (and *D*, diffusion coefficient) is not an intrinsic property to the material but rather a function of both the material and its environment. Stated differently, the $D_{h}$ and concomitant surface preference of an excipient can be strategically manipulated by tuning the feed solution properties and drying conditions; moreover the effects of fine-tuned parameters on the $D_{h}$ of an excipient can be screened prior to drying by DLS.

In addition to excipient $D_{h}$, we investigated excipient ζ as a possible predictive measure of preferential surface adsorption. The excipient ζ was manipulated both directly by functionalising the surfaces of the silica nanoparticles and indirectly by tuning the pH and ionic strength of the solvent. The effect of directly functionalising the particle surface was studied by direct comparison of samples 5-0[med-OH], 5-0[med-NH${}_{2}$], and 5-0[med-Octyl], which contained medium-sized silica NP decorated with hydroxyl (unfunctionalised), aminopropyl, or octyl surface moieties, respectively. The ζ values for these excipients in buffer (pH 5.5, no CaCl${}_{2}$) ranged from −8.7 mV at minimum (unfunctionalised) to 19.9 mV at maximum (aminopropyl). The ζ for octyl functionalised silica nanoparticles showed insignificant difference from the unfunctionalised silica (−7.1 mV). The size of the octyl-functionalised particles however, was significantly larger than that of the unfunctionalised particles (111 nm vs. 40 nm), suggesting agglomeration of the nanoparticles induced by hydrophobic interactions between the alkyl side chains.

To understand the relationship between ζ and surface adsorption for these samples, the percent preferential surface adsorption was plotted against ζ; the trend in preferential surface adsorption roughly followed the trend in ζ at room temperature (Figure 7).

Despite these initial results, more rigorous analysis of this relationship across all thirteen samples revealed no discernible correlation between excipient ζ and preferential adsorption at the particle surface (Figure A10). The relationship between the absolute value of ZP, the magnitude of electrostatic repulsion between particles, and excipient surface adsorption was also found to be ambiguous (Figure A11). It can therefore be concluded that the ζ of an excipient does not significantly influence its extent of enrichment at the particle surface within the range of ζ values studied (−13 to 26 mV) (Table 1 and Table 2). It should, however, be noted that this range of ζ values was relatively narrow (largely due to buffer shielding effects) and the lack of a relationship between ζ and surface enrichment may therefore be attributable to insignificant difference the ζ values compared.

Finally, we investigated the ability of the excipient to competitively displace the protein at the droplet surface. To achieve this, we plotted the enrichment of the protein at the particle surface against that of the excipient (Figure 8). Initially, we observed no significant correlation between the two ($R^{2}$ = 0.28). Limiting the dataset to only samples with excipient enriched surfaces, however, revealed a striking improvement in the correlation between excipient and protein surface enrichment; as hypothesised, a strong, inverse correlation ($R^{2}$ = 0.95) was found to describe the relationship between the two features. From this data it may be reasonably concluded that excipient preferential adsorption competes with that of the protein; the stronger the excipient’s preference for the air-droplet interface, the more protein-depleted the interface becomes. When the excipient demonstrates a preference for the droplet interior, however, the protein surface adsorption is not determined by competitive adsorption from the excipient but other factors.

## 3. Conclusions and Outlook

Work towards the development of a scalable platform for spray drying of core-shell structures with labile cargo was presented here. Our proposed approach circumvents industrially-undesirable emulsion methods and complicated drying techniques, demonstrating that the surface affinity of a biologic can be curbed by tuning the preferential surface adsorption of excipients in the feed solution. We validate this approach by showing that positive preferential adsorption by the excipient competitively displaces protein from the air-droplet interface and in turn, dried particle surface (Figure 8).

Our results suggest that the hydrodynamic diameter of an excipient $D_{h}$ can be used to predict the degree to which it adsorbs at the air-droplet interface; excipients with higher $D_{h}$ showed higher enrichment at the particle surface. The $D_{h}$ of a nanoparticle excipient could be tuned through buffer properties (ionic strength, pH) as well as the particle surface functionality and size of the Si$O_{2}$ core. Interestingly, the core size of the SiNP did not always predictably alter the $D_{h}$ under the relevant buffered conditions (in contrast to water) (Table 2). Rather, it seemed to be the degree of aggregation amongst SiNP under the spray drying conditions that most significantly influenced the in situ $D_{h}$. In fact, it was shown that measuring $D_{h}$ under conditions that best simulated the drying process (i.e., T = $T_{oulet}$) marginally improved the predictive power of $D_{h}$, increasing the correlation with excipient preferential adsorption (%) from $R^{2}$ = 0.59 to 0.64 (Figure 6). From these results it is clear that the utility of predictive parameters in spray dried systems depends not solely on the behaviour or properties of components in isolation, but rather on the behaviour of these components in the context of the whole system and its associated conditions.

Unlike $D_{h}$, the ζ of excipients did not predict their preference to localise at the particle surface (Figure A10). This suggests that excipient chemistry could be altered solely for the purpose of modifying aggregation (and in turn, surface adsorption) without introducing confounding effects from changes in ζ. This conclusion, however, is bound by the scope of this study; excipient zeta potentials varied only narrowly from −13 to 26 mV. Future studies investigating the influence of a broader range of ζ values could be useful to increase the generalisability of these findings.

Given the strong predictability of the $D_{h}$ parameter in determining excipient surface preference, we propose the use of the Trojan horse principle for controlled core-shell assembly by spray drying as depicted in Figure 9. By covalently tethering or non-covalently adsorbing low $D_{h}$ excipients at the surface of high $D_{h}$ nanoparticles, one can effectively ’hitch-hike’ the secondary component to the spray dried particle surface. The nanoparticle thus serves as a Trojan horse; the secondary component effectively assumes the large $D_{h}$ of the nanoparticle, resulting in surface enrichment as predicted by the diffusion theory of core-shell self assembly (Figure 1). By this approach, not only can one create preference for a core-shell architecture wherein the labile biologic is effectively encapsulated (and in turn, protected), but furthermore, one can control the chemical composition of the shell without being limited by the intrinsic properties of the isolated excipient. As such, this strategy is amenable to applications wherein it is desirable to introduce a specific molecular entity or functionality (e.g., gas/solvent permeability, wettability, targeting, etc.) to the particle surface (in contrast to those where the main aim is simply to limit protein surface adsorption).

In conclusion, this paper systematically investigates the relationship between colloidal properties of nanoparticle excipients in protein-containing feed solution and their relative enrichment at the surface of the spray dried particle. The hydrodynamic size, $D_{h}$ of the nanoparticle excipients studied was a clear predictor of their surface enrichment. On the other hand, ζ was not indicative of excipient surface representation within the obtained dry material. The use of high $D_{h}$ nanoparticles is shown to be a viable strategy for limiting protein adsorption at the air-droplet interface in single feed aqueous spray drying. Finally, a platform approach employing high $D_{h}$ nanoparticles as ’Trojan’ horses to carry low $D_{h}$ excipients to the droplet air interface (and surface of the dried particle) is proposed.

## 4. Materials and Methods

All materials were purchased from Merck/Sigma-Aldrich (Damstadt, Germany) unless otherwise specified. Semi-pure phytase was kindly gifted by AB Enzymes (Darmstadt, Germany).

### 4.1. Nanoparticle (NP) Functionalisation

#### 4.1.1. Seed-Growth Synthesis of Silica NP

Silica nanoparticles of 96 nm were synthesised via a seeding method using Ludox AS40 as the starting material. To 4.73 mL MilliQ water, 3.67 mL of 30% ammonium hydroxide solution was added slowly with stirring (400 rpm). To this, 0.226 g Ludox AS40 suspension was added. Finally, 2.9 mL tetraethyl orthosilicate (TEOS) was added to round bottom flask via a syringe pump at the rate of 0.2 mL/h. A 19G needle was necessary to resist clogging. The reaction was allowed to proceed for 12 h before centrifuging in water to remove residual ammonium hydroxide and TEOS (5× 13,000 RPM 4 °C, 30 min).

#### 4.1.2. Solvent Exchange of Ludox Silica

A solvent exchange was performed to redisperse ludox silica nanoparticles AS40 (’medium-sized’, $D_{h,water}$ = 16 ± 1 nm) in ETOH. LUDOX NP solutions were diluted (5×) in de-ionised water and centrifuged at 13,000 RPM (4 °C) for 30 min. At this point, a sedimented pellet (clear gel) was collected and redispersed in ethanol, washed another two times under the same conditions (13,000 RPM, 4 °C) and finally diluted in ETOH to achieve a final concentration of roughly 40 mg/mL.

#### 4.1.3. Aminopropyl Functionalised Silica NP

Aminopropyl functionalised SiNPs were obtained via an adapted literature procedure [24]. To 120 mL of Ludox AS40 redispersed in Ethanol was added 10 mL of APTES drop-wise. A plastic round bottom flask was used to avoid functionalisation of the glass surface. The solution was refluxed at 80 °C for 80 min, allowed to cool and subsequently centrifuged for 30 min at 13,000 RPM and 4 °C to remove unreacted APTES. Finally, the particles were dialysed against de-ionised water using a membrane with an 8000 g/mol molecular weight cutoff for a period of two days. Samples were analysed by DLS and ζ prior to spray drying.

#### 4.1.4. Octyl Functionalised Silica NP

Octyl functionalised SiNPs were obtained via an adapted literature procedure [24]. To a plastic round bottom flask containing 10 mL of Ludox AS40 redispersed in ethanol was added with 1.6 mL Triethoxy(octyl)silane. The solution was refluxed under Nitrogen at 85 °C for 80 min. The solution was allowed to cool and subsequently centrifuged for 30 min at 13,000 RPM and 4 °C to remove unreacted triethoxy(octyl)silane. Finally, the particles were dialysed against de-ionised water using a membrane with an 8000 g/mol molecular weight cutoff for a period of 2 days. Samples were analysed by DLS and ζ prior to spray drying.

### 4.2. Spray Drying

All spray drying was conducted on a BUCHI Mini Spray Dryer B-290 fit with a small cyclone. The inlet temperature was consistently between 137–138 °C and the pump was kept at 10% for all runs. The measured outlet temperature varied from 71 to 77 °C, with the mean temperature being 75 °C across all runs. The system was cleaned extensively between each run to prevent cross contamination.

### 4.3. X-ray Photoelectron Spectroscopy

XPS was used to determine the weight percent of silicon and sulphur elements on the surface of the spray dried particles (ca. 5 nm depth). Measurements were obtained using an Escalab 250Xi XPS instrument (Thermo Scientific, Waltham, MA, USA). Samples were prepared by mounting on double-sided copper tape.

### 4.4. Elemental Analysis

Bulk compositional analysis was performed by elemental analysis. The relative abundances of sulphur (S) and silicon (Si) in the spray dried powder were measured by inductively coupled plasma optical emission spectroscopy (ICP-OES).

### 4.5. Field Emission Gun Scanning Electron Microscopy

Particle morphology was characterised by SEM using a TESCAN MIRA3 FEG-SEM. Samples were prepared by direct deposition of freeze-dried powder on black carbon adhesive. The deposited sample was coated with Pt using the Quorum Technologies Q150T ES Turbo-Pumpted Sputter coater prior to imaging. Spray dried particle size analysis was performed over 200 particles per sample, using the Fiji open-source image-processing package, ImageJ software [25], version 1.53b, and Origin Pro 2018 software, version b9.5.1.195.

### 4.6. Dynamic Light Scattering and Zeta Potential

Dynamic light scattering (DLS) and zeta potential (ZP) measurements were performed using a Malvern Pananalytical Zetasizer Nano ZS90 instrument fitted with a He-Ne laser (λ = 663 nm). Samples measured in feed buffer media were measured at the concentrations relevant to the spray drying process. Measurements performed in water were made at ca. 0.1 mg/mL.

## Data Availability

The data presented in this study are available on request from the corresponding author.

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
