# Peer review of "A Platform Approach to Protein Encapsulates with Controllable Surface Chemistry"

_molecules, 2022, doi:10.3390/molecules27072197_

Round 1
Reviewer 1 Report
Scherman and co-workers systematically investigated the relationship between colloidal properties of nanoparticle excipients in protein-containing feed solution and their relative enrichment at the surface of the spray dried particle. The hydrodynamic size of the nanoparticle excipients was identified as a clear predictor of their surface enrichment, while ζ was not indicative of excipient surface representation within the obtained dry material. The use of high Dh nanoparticles is shown to be a viable strategy for limiting protein adsorption at the air-droplet interface in single feed aqueous spray drying. the also proposed a platform approach employing high Dh nanoparticles as ‘Trojan’ horses to carry low Dh excipients to the droplet air interface. This work is interesting and this paper is well organized, I would like to recommend its publication after minor revision:
- In the figure 3 legend, "Scale bar is 5 m for all micrographs " should be " Scale bar is 5 µm for all micrographs "
- error bat in figure A3 should be provided as in figure A2(a).
Author Response
As requested, we have made the following changes to our manuscript:
- Changed ‘5m’ to ‘5μm’ in the caption of Figure 3.
- Added error bars to the zeta potential measurements in Figure A3.
For a more extended list of changes, please also see the attached file.

Reviewer 2 Report
The study aims to develop a scalable platform for the fabrication of core-shell protein encapsulates by simple, single-feed spray drying to be scalable to an industrial level. The paper is well written, and the results are well presented. However, some improvements have to be made. In particular, it has been reported as the depth to which each factor was investigated was determined by the relative importance to the study, Key factors (excipient size, surface functionality, pH, [CaCl2]) were studied at three levels and secondary factors were studied only at two levels. Furthermore, Parameter levels were defined taking into consideration synthetic feasibility/commercial availability, protein stability, and industrial relevance. I suggest the authors improve the description of the criteria adopted to select these parameters because the way it is expressed seems arbitrary.
Author Response
As requested by the Reviewer, we have added the following text to improve the explanation for our parameter selection:
- “Feed solution parameters controlled for the purpose of direct or indirect tuning of excipient colloidal properties (i.e. nanoparticle core size, surface functionality, pH, [CaCl2]) were investigated at three levels (minimum of three samples compared). Two additional factors unrelated to excipient properties (total concentration of dissolved solids in the feed buffer ([Excipient+ Protein]) and excipient loading ratio ([Excipient]/[Excipient + Protein])) were also studied to understand their effect on particle morphology. Parameter levels are discretely defined in Figure 2.”
- “Figure 2. Formulation parameters investigated. Nanoparticle size and surface functionality were studied to probe the effect of directly modifying excipient molecular size and hydrophilicity. In contrast, feed solution pH and salt (CaCl2) concentration were investigated as indirect methods of controlling the effective Dh and colloidal stability of the excipient. The excipient loading ratio ([E]/([E]+[P])) and total dissolved solids content ([Solids]) were additionally studied to understand their impact on particle morphology. Parameter levels were defined taking into consideration synthetic feasibility/commercial availability, protein stability, and industrial relevance.”
For a more extended list of changes, please also see the attached file.
